# Improvement of Qualitative Analyses of Aliphatic Alcohols Using Direct Catalytic Fuel Cell and Chemometric Analysis Format

**DOI:** 10.3390/s24103209

**Published:** 2024-05-18

**Authors:** Mauro Tomassetti, Federico Marini, Riccardo Pezzilli, Mauro Castrucci, Corrado Di Natale, Luigi Campanella

**Affiliations:** 1Department of Chemistry, University of Rome, “La Sapienza”, P.le A. Moro 5, 00185 Rome, Italy; mauro.castrucci@libero.it (M.C.); luigi.campanella@fondazione.uniroma1.it (L.C.); 2Department of Industrial Engineering, University of Rome “Tor Vergata”, Via del Politecnico 1, 00133 Rome, Italy; riccardo.pezzilli@gmail.com; 3Department of Electronic Engineering, University of Rome “Tor Vergata”, Via del Politecnico 1, 00133 Rome, Italy; dinatale@uniroma2.it

**Keywords:** qualitative–quantitative analysis, aliphatic alcohols, direct catalytic methanol fuel cell, chemometrics, data fusion methods

## Abstract

Direct catalytic methanol fuel cells (DCMFCs) have been studied for several years for energy conversion. Less extensive is the investigation of their analytical properties. In this paper, we demonstrate that the behavior of both the discharge and charger curves of DCMFCs depends on the chemical composition of the solution injected in the fuel cell. Their discharge and charge curves, analyzed using a chemometric data fusion method named ComDim, enable the identification of various types of aliphatic alcohols diluted in water. The results also show that the identification of alcohols can be obtained from the first portion of the discharge and charge curves. To this end, the curves have been described by a set of features related to the slope and intercept of the initial portion of the curves. The ComDim analysis of this set of features shows that the identification of alcohols can be obtained in a time that is about thirty times shorter than the time taken to achieve steady-state voltage.

## 1. Introduction

Fuel cells have been mostly investigated for energy conversion applications [1,2]; however, fuel cells such as those making use of methanol or ethanol also shown noteworthy analytical properties [3,4].

For instance, fuel cells (DCMFCs) have been demonstrated to be suitable for quantifying the ethanol in beverages, such as wine or beer and other alcoholic drinks [5,6], with a precision and accuracy often better than those of standard analytical methods, but in a simpler, quicker and less expensive way. Besides beverages, the quantitative detection of ethanol has also been demonstrated in pharmaceutical products [7] and biological fluids, opening up the possibility of forensic applications [7]. Interestingly, molecules characterized by alcohol groups can also be detected by fuel cells; this feature has been tested on antibiotics such as chloramphenicol and imipenem, and on carbohydrates [7,8].

At this point we asked ourselves whether it was possible to use this type of fuel cell to carry out the analysis of different types of organic molecules for the qualitative recognition of different substances as well, and not only their quantitative analysis, especially for molecules of pharmaceutical or biological interest, as well as food. This is due to the fact that this kind of fuel cell provides a response to any organic substance that possesses one very alcoholic group or at least a group with partially alcoholic properties, though they will obviously have different response intensities depending on the number and the more or less alcoholic characteristics of the group (-OH) that characterizes them. It is clear, however, that, for correctly and simply observing the difference in responses, a method was needed that can comprehensively highlight the response of the fuel cell. We identified this method as the application of a chemometric analysis. This, indeed, has been demonstrated by us in recent research published in *Sensors* [9]. In that paper we highlighted that, if the different substances to be qualitatively recognized have an (-OH) group with quite different alcoholic properties, it was sufficient, in that case, to use an appropriate chemometric method [9] to obtain good separations in the “scores” values, which were obtained from all the data of the single discharge curves of the aforementioned type of fuel cell. We have, indeed, experimentally verified that this format is able to carry out valid qualitative recognitions of any unknown substances with one at least partially alcoholic group [9].

In summary, it was demonstrated that the qualitative (as well as obviously quantitative) recognition of different organic substances with at least one alcoholic group is possible when using only a direct catalytic methanol fuel cell (DCMFC) (Figure 1) simultaneously with an appropriate chemometric processing method for the experimental discharge curve data.

However, if the substances to be recognized are very chemically similar, such as aliphatic alcohols (mono-alcohols with only linear aliphatic chains, or with modest ramifications), it becomes much more difficult to obtain good separations, even when using chemometric methods. This is due to the “chemical” similarity of the different alcoholic molecules and, therefore, the probable similarity of their subsequent oxidative processes, which occur until H_2_O and CO_2_ are produced and take place in the catalytic fuel cell. In fact, this kind of fuel cell, which generally uses, as catalysts in its anodic section, classical catalysts based on platinum and ruthenium, generally work according to the same oxidative processes [5].

Despite this, we have now tried to overcome this difficulty, since we realized that if, in addition to the appropriate chemometric methods, exploiting data fusion strategies were applied to distinguish, and, therefore, recognize, the different aliphatic alcohols under consideration, through recording and processing, they could recognize the charge curves in addition to the discharge curves of the catalytic fuel cell. In this regard, we studied six different aliphatic alcohols, for which we recorded not only the usual discharge curves but also their corresponding charge curves, before verifying, with the appropriate chemometric method, the possibility of exploiting data fusion strategies [10] able to distinguish and then recognize the six different aliphatic alcohols considered.

## 2. Materials and Methods

### 2.1. Hydroalcoholic Solutions

The six investigated aliphatic alcohols (methanol, ethanol, iso-propanol, propanol, iso-amyl, sec-butanol) were of analytical grade and purchased from Sigma-Aldrich (St. Louis, MO, USA). All the hydroalcoholic solutions which were necessary for recording the discharging or charging curves of the fuel cell were obtained by dissolving the weighed quantity of each alcohol directly in the proper measured volume of distilled and deionized water (final conductivity 0.01–0.02 µS). Using these hydroalcoholic solutions, both the potentiostatic discharge curves (as µA vs. time) and the open-circuit fuel cell loading curves (as voltage vs. time) were recorded for the same fixed concentration of each alcohol.

### 2.2. Apparatus and Measurements

A H-TEC Model F111 DCMFC (50 × 50 × 40 mm) weighing 100 g, obtained from the Fuel Cell Store (College Station, TX, USA), was used. The cell was made of Plexiglas©, while the electrode end plate was made of a Pt-Ru black catalyst combined with a Nafion™ membrane. For potentiostatic format measurements, a Palmsens mod. EmStat potentiostat (Houten, The Netherlands) was used, which was connected to the fuel cell with a PSTrace Software, ver. 4.6, data interface, supplied by a Compaq Presario PC.

For each discharge-format potentiostatic measurement, after an hydroalcoholic solution (about 2 mL) with a fixed alcoholic concentration was introduced into the fuel cell, the supplied current was recorded and read point-by-point up to a steady state. The fuel cell was connected to EmStat using the anode as the working electrode, while the cathode acted as the reference and counter electrode. Before the current discharging measurement, the EmStat automatically measured the Open-Circuit Voltage (OCV) value for a time of about 200 s, and then the anode potential was set to the value of the Optimized Applied Potential (OAP = OCV − 100 mV) [5].

A set of charging measurements was also carried out by the fuel cell, which was filled with each studied hydroalcoholic solution and operated in batch mode. For each measurement, after a hydroalcoholic solution (about 2 mL) with a fixed alcoholic concentration was introduced into the fuel cell, the increase in the potential generated between the respective anodic and cathodic ends of the fuel cell was recorded up to a steady state (taking potential readings every 0.1 s) using the PalmSens, operating, in this case, in an open-circuit mode. The fuel cell operating in an open circuit and in batch mode recorded, in this case, the increase in the voltage charge curves. For every measurement, the voltage value was immediately and continuously recorded for each alcohol, point by point, until a steady state.

Once each of the discharging and charging curves were recorded, in both cases, at this point, the slope and the intercept value of the tangent line at the point of the maximum slope of every obtained curve, whether descending or ascending, was also calculated so as to be able to compare the respective values of the slopes and intercepts of the straight tangent lines obtained for every one of the six evaluated aliphatic alcohols.

### 2.3. Chemometric Methods

All experimental obtained data have been processed via a multi-block exploratory analysis, using in-house routines written in Matlab. More specifically, the Common Component and Specific Weight Analysis (CCSWA), also known as the Common Dimensions (ComDim) algorithm [10,11], was used.

ComDim can be considered a generalization of principal component analysis (PCA) [12,13,14,15] for cases where multiple data matrices are collected to describe the same set of samples. It was first applied in sensory analysis and has gained some popularity in both sensorimetry and chemometrics [10,11,16,17,18].

ComDim operates by extracting the components which summarize the maximum amount of common information among the different blocks of data under investigation. In practice, ComDim aims to determine the underlying components, together with the component-specific block weights (or saliencies), that reflect the importance that the various blocks of variables attributed to these components. In fact, using ComDim, a single set of scores can be extracted from all the data blocks considered. After that, given the scores, loadings are calculated for each matrix which can then be used for the interpretation of the observed variance.

Briefly [10,11,16,17,18], we consider a set of M data matrixes Xm m = 1…M sharing the same number of samples n, but possibly with a different number of variables. Each matrix is assumed to be mean-centered and further scaled to unit Frobenius’ norm, so to make the variance of the different blocks comparable. Then, by defining, for each block, the cross-product matrix $W_{m}=X_{m}X_{m}^{T}$, ComDim components are calculated based on the following iterative procedure.

For each component j:


Initialize the saliences: $\lambda_{j}^{\left(m\right)}=1\;\;\;\forall m=1,\cdots ,M$Estimate $q_{j}$ as the first left singular vector of $W=\sum_{m=1}^{M}\lambda_{j}^{\left(m\right)}W_{m}$Update the saliences as $\lambda_{j}^{\left(m\right)}=q_{j}^{T}W_{m}q_{j}$Repeat steps 2–3 until convergence.Deflate each block: $X_{m}^{\left(j+1\right)}=X_{m}^{\left(j\right)}-q_{j}q_{j}^{T}X_{m}^{\left(j\right)}⟹W_{m}^{\left(j+1\right)}=W_{m}^{\left(j\right)}-\lambda_{j}^{\left(m\right)}q_{j}q_{j}^{T}$Extract successive components by repeating steps 1–5 on $W_{m}^{\left(j+1\right)}$


The F-extracted common components (CCs) are then collected in the scores’ matrix $Q=[q_{1}\;\;\;\;q_{2}\;\;\;\;\cdots \;\;\;\;q_{F}]$, so that, for each block, the corresponding loadings can be calculated as $P_{m}=X_{m}^{T}Q$.

## 3. Results

To prove the assumption of this research, the usual discharge curves were constructed (and repeated at least three times) for each studied alcohol by recording the current of hydroalcoholic solutions with the same concentration until they reached a steady state. The voltage charge curves of each of the six studied alcohols were then also constructed, recorded, and repeated at least three times in fuel cells operating, of course, in this case, in an open-circuit mode, using hydroalcoholic solutions with the same concentration. Each experimental curve thus obtained is shown in Figure 2 and Figure 3, respectively, while their numerical files are reported in Appendix A.

The possibility of exploiting data fusion strategies, by means of ComDim in particular [10] (a multi-block generalization of principal component analysis), to simultaneously process the resulting data matrices derived from both the discharging and charging process curves (in Figure 2 and Figure 3) of six hydroalcoholic solutions made of different hydroalcoholic aliphatic alcohols, with the same concentrations, placed successively into a fuel cell made it possible to obtain a good separation even if the six studied alcohols were not very different, as they were all aliphatic (see Figure 4).

Figure 5 shows the loadings of the discharge and charge blocks along CC1 and CC2, displayed as line plots, as a function of time.

It can be observed how, in the case of the discharge curves, both the CC1 and CC2 curves have a continuous downward trend over time, very closely mirroring the downward trend—initially rapid, and then very slow, until a stabilization of their values—of the original discharging curves in Figure 2. On the other hand, the charging curves have more complex trends as a function of time, which also reflect the more varied trends of the values of the charging curves shown in Figure 3. This confirms the greater level of information in the charging curves, compared to the discharging curves, provided by the fuel cell.

Although the results of this method were undoubtedly accurate, a certain inconvenience manifested itself in the inevitable lengthening of time of the total process of the analysis. In fact, in this case, as has been said, the method involves recording not only each of the discharge curves, but also each of the charge curves, of each of the individual alcohols. These charge latter curves, although seeming to contain more information than the discharge ones, take a rather long time to be recorded (about an hour, or even more), since it is well known that the total oxidative process proceeds through various steps with set periods [2,3,19,20,21,22,23], which create a significant increase in total analysis times.

Therefore, we next investigated whether there was the possibility of overcoming this problem too, since we noticed that the trends of the curves, both for the discharging and even more so for the charging process, are characterized by a very fast initial trend, which is then followed by a rather slow process until reaching almost constant values (observe the typical curves of the respective discharging and charging processes in Figure 2 and Figure 3). We realized that, if the most rectilinear parts, both descending and ascending, of the respective discharging and charging curves of the different alcohols studied were selected, by calculating the equations of the straight lines of the tangents and relating these to the straighter parts of these curves, and the ComDim method was applied [10,18] once again, but only to the parameters of the equations of the straight tangent lines obtained (and reported in Table 1 and Table 2), it would easy to arrive at a much simpler data matrix. From this new matrix, it was possible to obtain, again by means of the ComDim algorithm, a sufficiently acceptable separation of the “scores” most representative of the different alcohols, as is shown in Figure 6, and we especially observed how this result could be obtained by using at least one-thirtieth of the experimental time necessary to record the entire charge and discharge curves of the total charging and discharging processes of the fuel cell and then processing all the data reported in the Appendix A using chemometrics, as achieved previously.

Finally, the loadings of the fitting parameters, calculated from the equations of the tangents of the discharge and charge curves, of the first two common components are displayed in Figure 7.

An inspection of the loadings in Figure 7 suggests that all the fitting parameters are relevant to explaining the differences observed among the alcohols. Methanol is characterized by higher values for almost all the fitting parameters of its charge curve (except for the intercept) and higher values for the intercept and slope of its discharge curve. On the other hand, higher values of the intercept of the charge curve and higher values of the slope ratio and the slope and intercept STD characterize iso-propanol and propanol.

Finally, there is, naturally, the possibility of carrying out a quantitative analysis of each of the alcohols highlighted too, by constructing the usual respective calibration curves for each alcohol as a function of time, increasing by the time each alcohol’s concentration in the hydroalcoholic solutions (see Table 3), and then using the fuel cell response, i.e., the final point values of the discharge curves, as was possible in previous research [5,6,7,8].

## 4. Discussion and Conclusions

As is well known, there are many methods for the quantitative and qualitative analysis of alcoholic substances. Leaving aside the oldest chemical assays [24] or densitometric ones used, above all, for ethanol [25], there are quite a few instrumental methods for the determination of alcoholic substances, which are based on gas chromatography [26,27], liquid chromatography, HPLC and ion chromatography [28,29,30], GC-MS [31], and also on colorimetric [32] and spectrophotometric methods (UV-Vis) [33,34], or NIR [35,36,37], and so on. There is no shortage of sensorial methods either [38,39,40]. Each of these methods is generally used as an alternative to the others, mainly because it is better suited to certain types of matrices than others. The aim of the present work, therefore, was not to propose a method that intends to replace the classical methods reported above. Rather, it was to propose a new method for the qualitative analysis of different aliphatic alcohols, based on very modern technologies, which are also rather infrequently applied to problems of this type, such as particular fuel cells and classic [15] or very recent [10] chemometric procedures. The aim was to, therefore, let the reader know that this new analytical possibility also exists and has the advantages of being very simple, rapid, and, above all, of not requiring very expensive instrumental equipment.

We consider, therefore, the results obtained in this paper to be of remarkable interest, since, first of all, we realized that by recording and processing not only the discharging curves of a catalytic fuel cell but also its charging curves, using the same alcohols and, in addition, appropriate chemometric methods (i.e., the possibility of exploiting data fusion strategies), the fuel cell itself can easily distinguish and, therefore, recognize very similar aliphatic alcohols as well. Secondly, we demonstrated a drastic shortening of times, which actually makes the application of this type of qualitative analysis, if the most rectilinear parts, both descending and ascending, of the respective discharging and charging curves of the various alcohols are selected, then calculating the equations of the straight lines of the tangents relating to the straighter parts of these curves. By applying the ComDim fusion method [10], once again, the parameters of the equations of the tangents of the straight-line parts of the ascending and descending curves can be obtained. In brief, when working with this type of format, i.e., using only the fuel cell (DCMFC) and ComDim data fusion strategy, this proposed aliphatic alcohol recognition method becomes more practical and much more acceptable.

Finally, despite this paper having shown that DCMFC results, once analyzed using a chemometrics data treatment, are a valid and simple alternative format for the detection and identification of organic molecules characterized by at least one alcohol group, a likely further improvement of this format will be made possible by the integration, into the anodic compartment of the DCMFC device, of nanostructured materials, such as Layered Double Hydroxides (LDHs), also known as hydrotalcite-like compounds (or even enzymes such as catalase) [23].

## Figures and Tables

**Figure 1 sensors-24-03209-f001:**
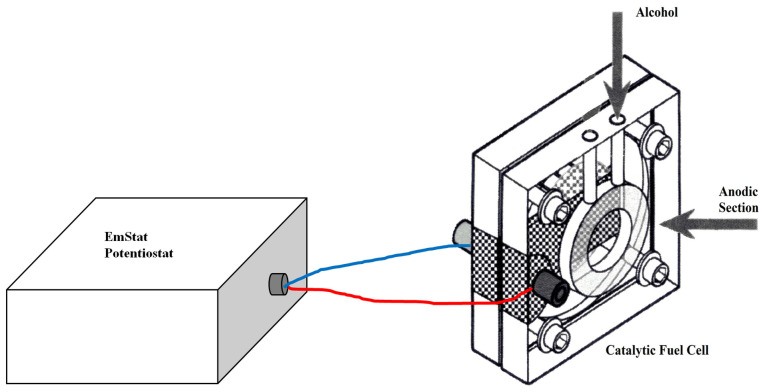
Representation of direct catalytic methanol fuel cell (DCMFC) used in this and previous research.

**Figure 2 sensors-24-03209-f002:**
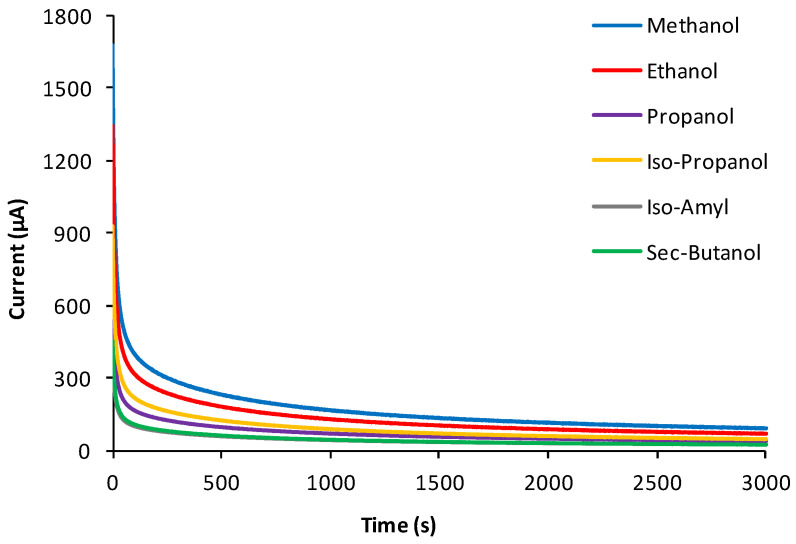
“Discharge curves”, i.e., the current discharging response trends of 5 × 10^−4^ mol L^−1^ aqueous solutions of each of the six investigated alcohols, all data points of which constitute the data set under investigation. Each curve represents the mean of at least three determinations.

**Figure 3 sensors-24-03209-f003:**
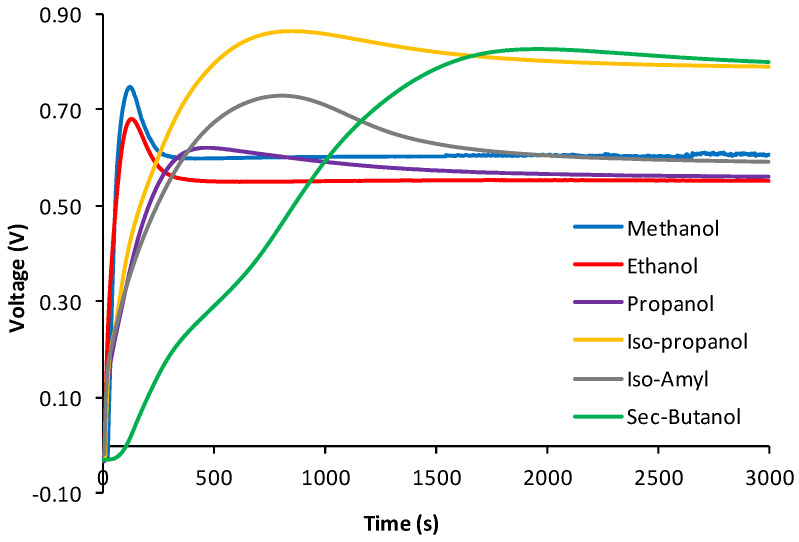
“Charge curves”, i.e., the voltage charging response trends of 5 × 10^−1^ mol L^−1^ aqueous solutions of the six alcohols under investigation, all data points of which also constitute the data set under investigation. Each curve represents the mean of at least three determinations.

**Figure 4 sensors-24-03209-f004:**
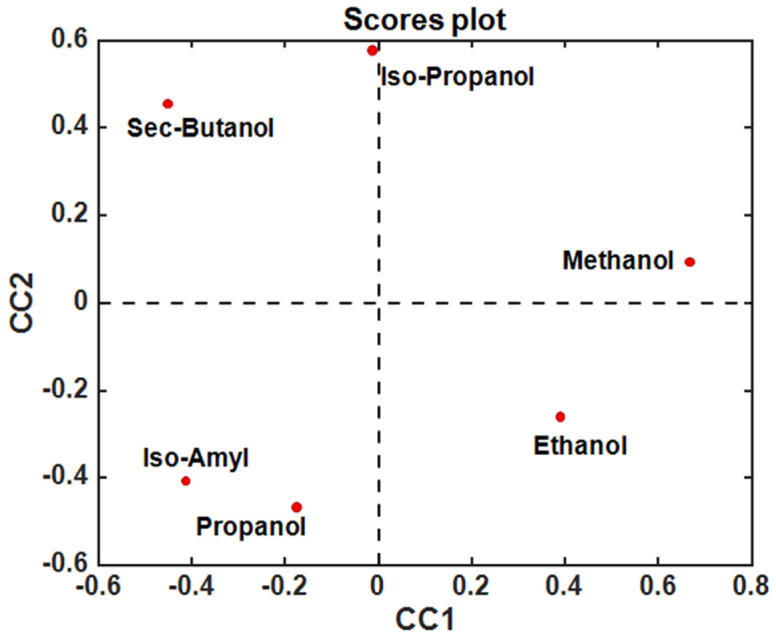
Score plot. Projection of the six analyzed alcohols onto the first two common components (CC1 and CC2) extracted by ComDim.

**Figure 5 sensors-24-03209-f005:**
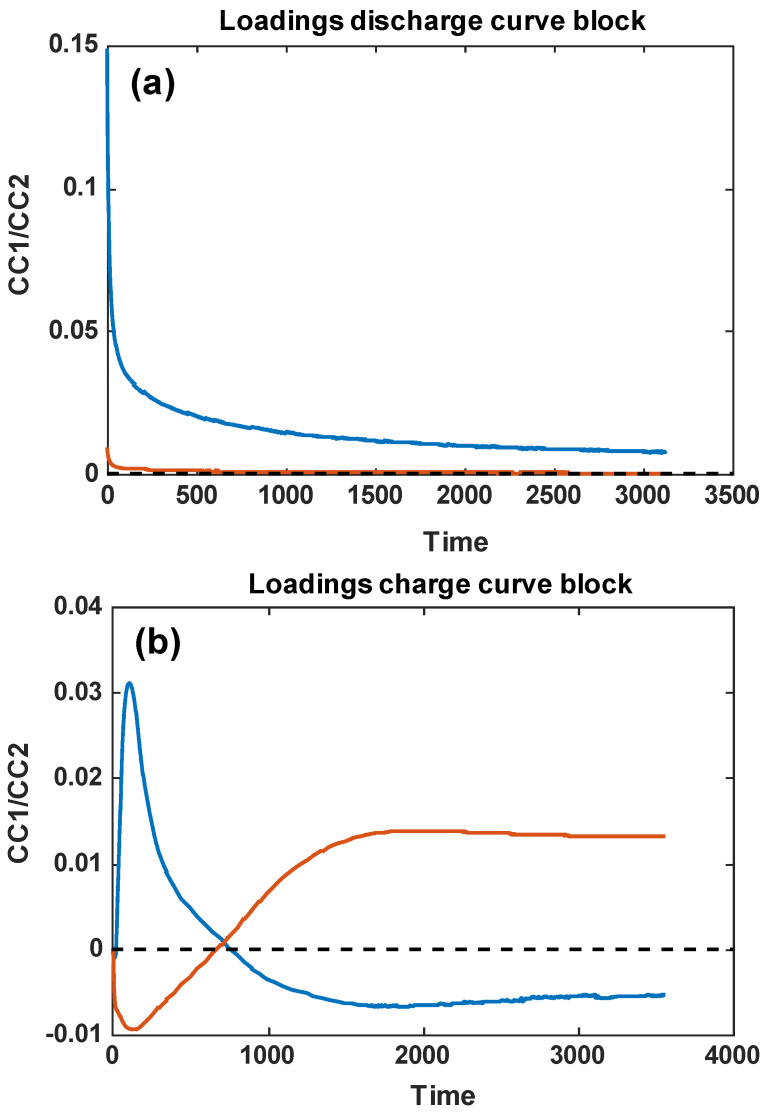
Loading projections of the six analyzed alcohols onto the first two common components, as extracted by ComDim. (**a**) Loading discharge curve block and (**b**) loading charge curve block. Blue lines correspond to CC1, while the orange ones correspond to CC2.

**Figure 6 sensors-24-03209-f006:**
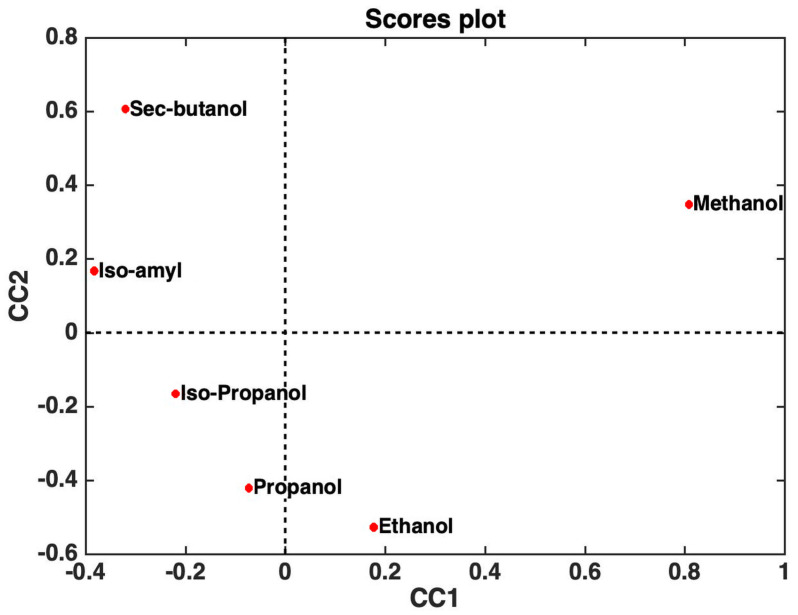
Projection of the six analyzed alcohols onto the first two common components (CC1 and CC2), extracted by a “multiblock” exploratory analysis of the data set, which resulted from fitting the parameters, reported in Table 1 and Table 2, of the straight-line equations of the tangents related to these straighter parts of the discharging and charging curves.

**Figure 7 sensors-24-03209-f007:**
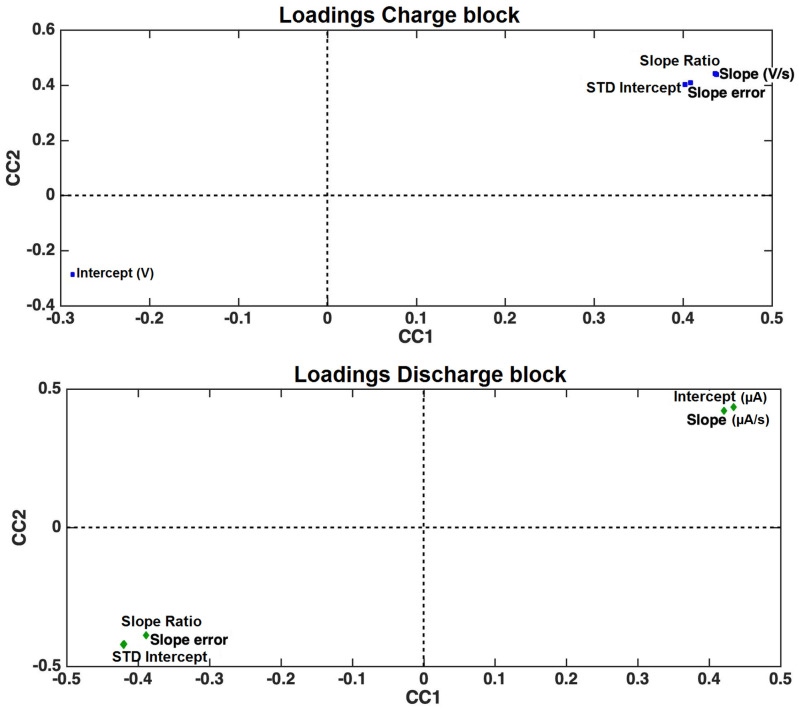
Representations of the loadings of the charge (upper panel) and discharge (lower panel) curves, obtained by applying ComDim to the data set created using only the data reported in Table 1 and Table 2.

**Table 1 sensors-24-03209-t001:** Values of the equation parameters of the tangent to the straight-line part of the discharge curves.

Alcohol Investigated	Slope (µA/s)	Slope SD	Intercept (µA)	Intercept SD	Slope Ratio %
**Methanol**	−125.3345	±10.64253	1683.88899	±13.75317	100.000
**Ethanol**	−175.4311	±29.98601	1468.58510	± 89.51645	139.9703194
**Iso-propanol**	−267.34291	±21.08728	1254.20004	±235.11949	213.3035278
**Propanol**	−320.98941	±28.08175	1129.06917	±320.16881	256.1061799
**Iso Amyl**	−381.46361	±37.75922	988.01255	±416.05487	304.3564302
**Sec-Butilic**	−373.47291	±36.43792	1006.65093	±403.38463	297.980931

**Table 2 sensors-24-03209-t002:** Values of the equation parameters of the tangent to the straight-line part of the charge curves.

Alcohols Investigated	Slope (V/s)	Slope SD	Intercept (V)	Intercept SD	Slope Ratio %
**Methanol**	0.01233	±2.09695 × 10^−4^	−0.23293	±1.21010 × 10^−2^	100.000
**Ethanol**	0.00548	±1.23101 × 10^−5^	−0.15746	±7.73909 × 10^−4^	44.4444444
**Iso-propanol**	0.00129	±4.62291 × 10^−6^	−0.27865	±9.35781 × 10^−4^	10.4622871
**Propanol**	0.00109	±2.34966 × 10^−6^	−0.27218	±5.39314 × 10^−4^	8.84022709
**Iso Amyl**	9.69E-04	±3.37833 × 10^−6^	−0.26091	±8.77158 × 10^−4^	7.85951338
**Sec-Butilic**	6.88E-04	±5.05715 × 10^−7^	−0.09273	±4.22026 × 10^−4^	5.58276561

**Table 3 sensors-24-03209-t003:** Straight-line regression equations of each alcohol vs. the increasing alcohol concentration used in the hydroalcoholic solutions.

	Regression Equation [y = ax + b]		Linearity Range
	a [µA/mol L^−1^]	b [µA]	[mol L^−1^]
**Methanol**	(21.8 ± 0.7) × 10^3^	75.9 ± 3.2	(1.0 × 10^−3–^8.0 × 10^−1^)
**Ethanol**	(17.8 ± 0.6) × 10^3^	61.1 ± 3.0	(1.0 × 10^−3^–8.0 × 10^−1^)
**Iso-propanol**	(4.27 ± 0.08) × 10^3^	47.0 ± 2.7	(8.5 × 10^−4^–8.0 × 10^−1^)
**Propanol**	(3.32 ± 0.1) × 10^3^	35.0 ± 1.6	(7.5 × 10^−4^–8.0 × 10^−1^)
**Iso-Amyl**	(2.21 ± 0.3) × 10^3^	20.1 ± 2.1	(6.5 × 10^−4^–8.0 × 10^−1^)
**Sec-Butanol**	(2.43 ± 0.4) × 10^3^	22.4 ± 1.9	(6.5 × 10^−4^–8.0 × 10^−1^)

## Data Availability

All experimental data are available in the Appendix A and in the text of this paper.

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
