# Peer review of "Improvement of Qualitative Analyses of Aliphatic Alcohols Using Direct Catalytic Fuel Cell and Chemometric Analysis Format"

_sensors, 2024, doi:10.3390/s24103209_

Round 1
Reviewer 1 Report
Comments and Suggestions for Authors
The authors of this paper used a chemometric data fusion method called ComDim, which allows the identification of different types of aliphatic alcohols diluted in water, to analyze the behavior of the discharge and charger curves of Direct Catalytic Methanol Fuel Cells (DCMFCs) depending on the chemical composition of the solution injected in the fuel cell.
The topic is original and relevant. The authors first considered whether Fuel Cell could be used to analyze different types of organic molecules for qualitative analyses and proved that if the different substances, to be qualitatively recognized, have an (-OH) group with quite different alcoholic properties, it was sufficient, in this case, the use of an appropriate chemometric methods to obtain good separations, so this format was to be able to carry out valid qualitative recognitions of any unknown substances, having one, at least partially alcoholic group.
However, the authors made an effort to address the challenge of successfully establishing a separation, even with chemometric techniques, which becomes more difficult if the molecules that need to be detected have extremely similar chemical properties, as is the case with aliphatic alcohols.
The reasons and evidence used to support the conclusions are consistent.
The citations are suitable.
The representations of the loadings of the charge (upper panel) and discharge (lower panel) curves in Figure 7, which were produced by applying ComDim on the dataset using only the information provided in Tables 1 and 2, seem superfluous and misleading to me.
Author Response
Referee (n. 1)
-Comments and Suggestions for Authors
The authors of this paper used a chemometric data fusion method called ComDim, which allows the identification of different types of aliphatic alcohols diluted in water, to analyze the behavior of the discharge and charger curves of Direct Catalytic Methanol Fuel Cells (DCMFCs) depending on the chemical composition of the solution injected in the fuel cell.
The topic is original and relevant. The authors first considered whether Fuel Cell could be used to analyze different types of organic molecules for qualitative analyses and proved that if the different substances, to be qualitatively recognized, have an (-OH) group with quite different alcoholic properties, it was sufficient, in this case, the use of an appropriate chemometric methods to obtain good separations, so this format was to be able to carry out valid qualitative recognitions of any unknown substances, having one, at least partially alcoholic group.
However, the authors made an effort to address the challenge of successfully establishing a separation, even with chemometric techniques, which becomes more difficult if the molecules that need to be detected have extremely similar chemical properties, as is the case with aliphatic alcohols.
The reasons and evidence used to support the conclusions are consistent.
The citations are suitable.
The representations of the loadings of the charge (upper panel) and discharge (lower panel) curves in Figure 7, which were produced by applying ComDim on the dataset using only the information provided in Tables 1 and 2, seem superfluous and misleading to me.
Response to the Referee (n. 1).
We thank Referee n.1, both for the lucid brief description of the objectives and innovations introduced to achieve these objectives, and above all for having appreciated these innovations. Unfortunately, we cannot agree with the request contained in the last sentence of the Referee, which suggests the suppression of the representations of the Loadings, shown in figure 7. We understand that these representations, especially if we do not consider the quadrants in which the scores fall, or if the reader is not very familiar with this type of representation of the Loadings, it may cause some misleading, which the Referee would like to avoid; however, precisely for these reasons, the possible deletion of what is reported in figure 7, would make it even more difficult to understand, even the comments reported in the text of the work, regarding the Loadings represented in Fig. 7.

Reviewer 2 Report
Comments and Suggestions for Authors
The topic of the present study and the obtained results are very interesting. However, low quality of the presentation and lack of the comparison with current methods shade the true value of the study.
1. What are the current methods used for aliphatic alcohols analysis? The advantages of the method proposed within the present study should be clearly pointed out.
2. I believe that more detailed experimental conditions should be given. I do not feel that I would be able to reproduce the experiment based on the data currently given in the manuscript. Some more details such as the dimension of the cell, membrane, catholyte/anolyte exact composition, electrode size, etc., could be given.
3. Figures 2 and 3 are blurry – their quality should be improved.
4. What is the limit of detection of aliphatic alcohols in the present setup?
5. More recent references should also be included in the manuscript.
6. There is a lot of formatting errors:
“For instance, Fuel Cells (DCMFC)…” should be “For instance, DIRECT CATALYTIC METHANOL Fuel Cells (DCMFC)…”
“in Sensors JournalS” should be “in Sensors Journal”
Why is Fuel Cell almost always spelled with capital letters?
The authors should check the manuscript for subscripts/superscripts.
Comments on the Quality of English Language
I would recommend thorough style revision. The Introduction contains very long sentences (one sentence = one paragraph) that are very difficult to follow/understand. Examples of these sentences are:
“At this point we asked ourselves….”
“In this paper we highlighted that,…”
“However, if the substances to be recognized…”
These long sentences should be split in smaller ones.
Author Response
Referee (n. 2).
Comments and Suggestions for Authors
The topic of the present study and the obtained results are very interesting. However, low quality of the presentation and lack of the comparison with current methods shade the true value of the study.
- What are the current methods used for aliphatic alcohols analysis? The advantages of the method proposed within the present study should be clearly pointed out.
- I believe that more detailed experimental conditions should be given. I do not feel that I would be able to reproduce the experiment based on the data currently given in the manuscript. Some more details such as the dimension of the cell, membrane, catholyte/anolyte exact composition, electrode size, etc., could be given.
- Figures 2 and 3 are blurry – their quality should be improved.
- What is the limit of detection of aliphatic alcohols in the present setup?
- More recent references should also be included in the manuscript.
- There is a lot of formatting errors:
“For instance, Fuel Cells (DCMFC)…” should be “For instance, DIRECT CATALYTIC METHANOL Fuel Cells (DCMFC)…”
“in Sensors JournalS” should be “in Sensors Journal”
Why is Fuel Cell almost always spelled with capital letters?
The authors should check the manuscript for subscripts/superscripts.
Comments on the Quality of English Language
I would recommend thorough style revision. The Introduction contains very long sentences (one sentence = one paragraph) that are very difficult to follow/understand. Examples of these sentences are:
“At this point we asked ourselves….”
“In this paper we highlighted that,…”
“However, if the substances to be recognized…”
These long sentences should be split in smaller ones.
Responses to the Referee (n. 2).
- First of all, we thank Referee n. 2, for having judged the topic and the results obtained from our work interesting.
Responses to individual questions asked by the Referee n. 2:
-(1) In the paragraph “Discussion and conclusions” of the revised paper, we briefly mentioned the most well-known classical instrumental methods, reported in the literature, for the determination of alcohols, also including a good number of new references, available in the literature, in this regard (see new Ref. 25-42, in the revised paper). These instrumental methods are so well known that we do not believe they should be described and commented further in this our article, not even compared with this method proposed by us, which in fact certainly does not intend to replace these well-known instrumental methods. In fact, our intent was above all to propose a further new method, the application of which, if nothing else, could also lead to the possibility of being able to determine qualitatively (but also quantitatively at the same time) an alcohol (building in addition a simple traditional calibration curve, at increasing hydroalcoholic concentrations (See Ref. 5, 6, of the paper)). The advantage therefore, in addition to that of speed of application, could also be represented by the simplicity, but above all the extremely limited cost, compared to other instrumental methods (such as for example HPLC, or spectroscopic ones, whose instrumentation is not cheap), given that instead, the commercial cost of a fuel cell, like the one we use, can also be in the order of less than 500 Euros, while that of an EmStat, for voltammetric and amperometric measurements (not of the most recent type), like the one from we used, it was (at the time) no more than 700-800 Euros.
-(2) We have added some further details in the experimental section, but without exaggerating, also because the description of the instrumentation and the measurement procedure have been reported in even more detail, including the optimization of the potentiostatic measurements themselves, in our works on fuel cells, already published, cited in the work (see for instance 5,7,8, 24). On the other hand, precisely in this regard, previously, other Referees had, on the contrary, criticized us for wanting to republish specifications and technical data, already widely reported in previous works.
Furthermore, some specifications (for example those of the concentrations of the hydroalcoholic solutions used) were reported, for convenience, in the captions of the figures to which they refer, so it was superfluous to also report them in the experimental section.
-(3) We have tried to give maximum definition to Figures 2 and 3, also improving their quality as much as possible. We hope that what we have done is sufficient.
-(4) In the current configuration the LOD, although slightly different for the individual alcohols tested, was however always of the order of (10-3 – 5x10-4) mol L-1, therefore in agreement with the linearity intervals reported in Table 3 of this work. But even these specifications can already be found in our previous works, cited in the text, especially for ethanol, or methanol.
-(5) Frankly, I don't really understand which "more recent" references the Referee is referring to, given that we frankly don't know of specific references on this same topic, i.e. (qualitative analysis using Fuel Cell and chemometric methods) published by other authors, neither recent nor previous. On the other hand, we were forced to remove at least three other references of ours in this regard from the text, as the Editor and the Editorial Board had previously peremptorily forced us to drastically reduce our self-citations. In any case, our most recent "Full Paper", in this regard, is the one cited in the work with n. 24 (apart from some possible "abstracts" of very recent conferences, which certainly do not need to be cited).
-(6) We did not invent the acronyms (DCMFC), or similar, but we adopted those already reported in the literature. Therefore, to make the meaning of the acronym extremely understandable, we have reported the meaning alongside and in full: Direct Catalytic Methanol Fuel Cell (DCMFC). It seems to us the simplest, most direct and understandable thing. Frankly, I don't quite understand how the Referee wanted us to report them, maintaining the same clarity.
-(7) We have corrected Sensors JournalS, removing the final “S”, we thank the Referee for reporting the oversight to us.
-(8) Why is Fuel Cell almost always written with a capital letter?
In reality there is no real reason, but it is a kind of "quirk" that we have allowed ourselves to highlight in the text what is the key word of the whole work, namely Fuel Cell. However, really if the Referee doesn't like this, we can also double-check the entire text by removing the capital letters wherever the wording Fuel Cell appears, obviously.
-(9) We have rechecked all the superscripts and subscripts in the manuscript and corrected some which, it is possible, have automatically moved, in the format changes (Word-pdf), which have occurred. However, if you still find something that is incorrect, please kindly report it to us, or briefly correct it yourself. Thank you.
-(10) Finally, as already mentioned in the previous resubmission, the text was rechecked by a native English-speaking professor. However, we tried to change the three sentences, which the Referee did not particularly like, as far as possible, but to which, frankly, the native speaker reviewer had not objected, judging them certainly not elegant, but at least sufficiently clear, which (as I hope the Referee too will agree), it is certainly the most important thing for a scientific paper.

Reviewer 3 Report
Comments and Suggestions for Authors
The manuscript sensors-2929821 presents how Direct Catalytic Methanol Fuel Cells can be a valid and simple alternative format for the detection and identification of organic molecules characterized by, at least, one alcohol group.
The work is fundamentally sound: the methodology was carefully exhaustive, incorporating a proper use of data fusion, and I found no fault whatsoever with the data analysis, or conclusions.
Author Response
Referee (n. 3).
Comments and Suggestions for Authors
The manuscript sensors-2929821 presents how Direct Catalytic Methanol Fuel Cells can be a valid and simple alternative format for the detection and identification of organic molecules characterized by, at least, one alcohol group.
The work is fundamentally sound: the methodology was carefully exhaustive, incorporating a proper use of data fusion, and I found no fault whatsoever with the data analysis, or conclusions.
Response to the Referee (n. 3).
We warmly thank Referee n. 3, especially for highlighting how Direct Catalytic Methanol Fuel Cells can be a valid and simple alternative format for the detection and identification of organic molecules characterized by, at least, one alcohol group. For also having judged the methodology to be exhaustive, and the conclusions to be completely correct.
We are truly happy that the Referee appreciated the novelty of the proposal of this new format, introduced by us.
